# MPC-Minimized Secure LLM Inference

## Abstract

Many inference services based on large language models (LLMs) pose a privacy concern, either revealing user prompts to the service or the proprietary weights to the user. Secure inference offers a solution to this problem through secure multi-party computation (MPC), however, it is still impractical for modern LLM workload due to the large overhead imposed by MPC. To address this overhead, we propose MARILL, a framework that adapts LLM fine-tuning to minimize MPC usage during secure inference. MARILL introduces high-level architectural changes during fine-tuning that significantly reduce the number of expensive operations needed within MPC during inference, by removing some and relocating others outside MPC without compromising security. As a result, MARILL-generated models are more efficient across all secure inference protocols and our approach complements MPC-friendly approximations for such operations. Compared to standard fine-tuning, MARILL results in $2.2-11.3\times$ better runtime and $2.4-6.9\times$ better communication during secure inference across various MPC settings, while typically preserving over 90% performance across downstream tasks. Anonymous code is available at `https://anonymous.4open.science/r/MPC-auto-B100`.

## 1 Introduction

Transformer-based large language models (LLMs) have revolutionized machine learning (ML). Since the announcement of ChatGPT, we have seen the release of a plethora of proprietary LLMs like GPT-4 (OpenAI, 2023), Claude 2 (Anthropic, 2024), and Gemini (Google, 2024), as well as open-source LLMs like Llama (Touvron et al., 2023) and Mistral (Jiang et al., 2023) that are now competitive against their proprietary counterparts (Chiang et al., 2024; Wang et al., 2023; Yan et al., 2024; Liu et al., 2024). Recently, companies have started to finetune these models on domain-specific data to improve their performance on downstream tasks such as chatbots, virtual assistants, and copilots (OpenAI, 2023; Anyscale, 2023; Cohere, 2024).

Using these finetuned models to power such user-facing services, however, raises significant privacy concerns. On one hand, the providers of these finetuned models do not want to expose their models' weights, as these models are often trained on proprietary data and represent competitive differentiation. On the other hand, users do not want to send their queries to these providers as these queries might contain sensitive or proprietary information (e.g. IP-protected code or user data). In fact, some enterprises prohibit their users from using LLM services, e.g., Samsung recently banned the use of external LLM services after an employee accidentally leaked sensitive code to ChatGPT (Ray, 2023).

Secure inference is a promising solution to address this challenge as it can provide privacy for both parties through secure multi-party computation (MPC) (Goldreich et al., 1987; Yao, 1986). There is a long line of work on MPC-based secure inference (Mohassel & Zhang, 2017; Mishra et al., 2020; Rathee et al., 2020; 2021; Wagh et al., 2019; Tan et al., 2021; Jawalkar et al., 2024) offering different performance and security tradeoffs, with the recent work focusing on secure transformer inference (Li et al., 2023b; Wang et al., 2022; Dong et al., 2023; Lu et al., 2025; Hou et al., 2023; Gupta et al., 2024). In principle, the service provider can use any of these recent secure inference protocols to support its privacy-preserving service. However, despite massive strides in efficiency, these protocols are still impractical for today's LLMs. For instance, the state-of-the-art solution (Gupta et al., 2024) requires 23 s and 15.9 GB of communication for the first token generation on a small 137M parameter model with 1024 input tokens. We expect the runtime and communication to degrade to around 6.5 minutes and 240 GB for a more typical 7B parameter model, which is impractical.

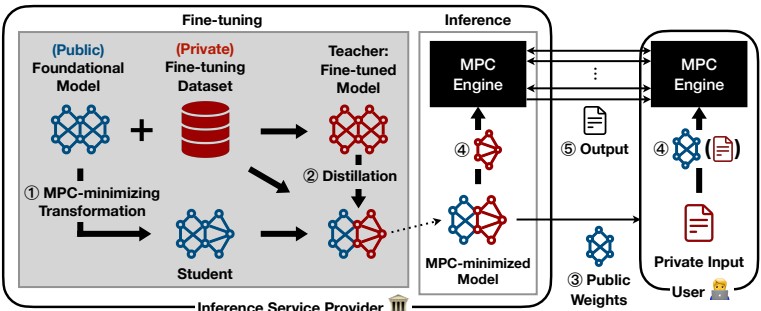

Figure 1: End-to-end workflow of our system. The private and public weights are highlighted in red and blue, respectively. The gray region represents our secure inference fine-tuning framework, MARILL, *run locally at the service provider* to output an MPC-minimized inference model. This model has public and private weights, and it enables secure inference (steps 3–5) that incurs high MPC costs only for private weights while maintaining security (§ 3). The provider only inputs the private part of the model to the MPC engine, and the user locally evaluates the public part of the model on its private input and feeds the partial inference result to the MPC engine. The model architecture for the private part is public and also input to the engine by the client, but it is omitted from the figure for simplicity. Single token generation is shown; subsequent tokens follow similarly since the client knows prior generated tokens. For clarity, the figure shows two parties running MPC engine instances, though some protocols include an additional helper party for faster secure inference (Appendix A).

To reduce this overhead, prior work has focused on expensive low-level operations within MPC and proposed *MPC-friendly* approximations for those operations (§ 2). In this work, we consider an orthogonal approach targeting high-level architectural changes, that offer a complementary way to minimize the MPC overhead. Instead of simplifying operations, such architectural changes reduce the number of expensive low-level operations needed within MPC. Importantly, this strategy does not (necessarily) eliminate these operations from the inference process entirely; rather, it relocates them outside of MPC without compromising security, where their cost is relatively negligible. Our work is the first to explore this *MPC-minimization* strategy. Notably, our strategy targets the fine-tuning phase, which is performed *locally* by the service provider before deploying the model through a secure inference API. The key insight is that *fine-tuning, when carefully tailored to secure inference, can unlock significant opportunities for MPC-minimization during inference*. Since our focus is on MPC-minimization, these fine-tuning changes do not accelerate plaintext inference (see Appendix C) and only serve to reduce secure inference costs.

Building on the above insight, we propose a secure inference fine-tuning framework MARILL[1], which relies on MPC-guided techniques that diverge from standard fine-tuning to improve the cost of secure inference. Models produced by MARILL are (i) MPC-minimized without compromising security (§ 3), and (ii) achieve ML performance close to that of standard fine-tuned models through knowledge distillation (§ 5). Crucially, since MARILL essentially compresses the model within MPC during inference, the resulting models are significantly more efficient across all secure inference protocols (§ 6.1). Furthermore, MARILL only introduces high-level architectural changes that complement MPC-friendly approximations. We show that integrating these approximations with MARILL leads to further efficiency improvements (§ 6.3).

MARILL builds on three techniques, all focused on the shared goal of minimizing a high-level model component within secure inference MPC via fine-tuning. The first two techniques adapt well-known fine-tuning methods to the MPC-based secure inference setting, representing a novel application of these ideas in this context. While these methods have previously also been used to accelerate secure machine learning in trusted execution environments (TEEs) (Zhang et al., 2024; Huang et al., 2024), their adaptation to MPC involves key distinctions due to the differences in the cost profile of MPC and TEEs, which we summarize in Appendix D.2. Our third technique is novel and specifically designed to optimize for the unique performance characteristics of MPC-based secure inference. Now, we present a brief overview of our techniques and the model component (underlined) they minimize:

---

[1]MARILL stands for MPC-Minimized ARchitecture for Secure Inference of LLMs

- **Leveraging open-sourced models**: As alluded to earlier, open-source LLMs have become more powerful and are now competitive against proprietary models (Chiang et al., 2024; Wang et al., 2023; Yan et al., 2024; Liu et al., 2024). Consequently, a trend has emerged where an increasing number of service providers opt to fine-tune these open-source models with their private datasets instead of pre-training their own proprietary models (Anyscale, 2023; Cohere, 2024). Since the open-source model weights are publicly available, they can be utilized to improve secure inference cost as they do not have to be kept private from the user. However, standard fine-tuning updates all the model weights with the private data, thereby necessitating that all weights be private and precluding any potential benefits of the open-source model weights. In light of this, we adapt two existing fine-tuning techniques to effectively leverage the public weights and minimize MPC:

    - **Layer Freezing** (§ 5.1): We reduce the number of transformer layers that need to be evaluated within MPC by restricting fine-tuning updates (and thus, private weights) to just the final layers of the pre-trained model. We resort to such strict demarcation because alternating private and public layers still require the bottleneck operations in the public layers to run within MPC (§ 4), and simply pruning the public layers leads to poor task performance (§ 6.4).
    - **Low-rank Adaptation (LoRA)** (§ 5.2): Recent parameter-efficient fine-tuning techniques like LoRA (Hu et al., 2022) have shown that it is possible to achieve comparable task performance by fine-tuning only a small fraction of the model's weights. Although LoRA was originally designed to reduce memory requirements during fine-tuning, we show that it can be repurposed such that the typical MPC cost of linear layers is incurred only for a smaller weight matrix dimensionality – a runtime bottleneck in the natural two-party setting as well as during decoding (Appendix B)-stages in other MPC settings (§ 5.2).

- **Modifying self-attention architecture**: We analyzed the cost profile of secure LLM inference under various MPC settings and identified self-attention as the bottleneck in the most efficient settings (§ 5.3). Fine-tuning can mitigate this overhead by enabling the pruning of certain heads in the (multi-head) self-attention architecture (Michel et al., 2019). However, to achieve significant improvements, we have to prune up to 75% heads (and their corresponding parameters) and this leads to a large accuracy drop despite fine-tuning (§ 6.4). To address this loss, we propose a novel head reduction technique that preserves accuracy with fine-tuning even for a large head reduction:

    - **Head-merging** (§ 5.3): We reduce the number of attention heads in self-attention by merging $m$ heads into one, but simultaneously, we also increase the head dimension proportionally to preserve all the pre-trained parameters. While it seems that we did not gain anything because the computational FLOPs remain the same, we show that head-merging actually matches the secure inference cost of head-pruning (§ 6.4). This is based on the key observation that the self-attention operations that are the bottleneck in MPC only scale with number of heads and not the head dimension. Our experiments show that if the heads are merged carefully, head-merging achieves much better accuracy than head-pruning since it preserves all the pre-trained parameters (§ 6.4).

The end-to-end workflow of MARILL is summarized in Fig. 1. Compared to standard fine-tuning, MARILL-generated models have $2.2 - 11.3\times$ faster runtime and $2.4 - 6.9\times$ lower communication across state-of-the-art secure inference frameworks in various MPC settings (§ 6.1). We evaluate the ML performance of MARILL on three different kinds of tasks, namely, code generation (Chen et al., 2021), chatbot (Zheng et al., 2023), and machine translation (Kocmi et al., 2022). Across these benchmarks, we show that MARILL typically preserves over 90% of the standard fine-tuned performance (§ 6.2).

## 2 RELATED WORK

**Secure Inference Protocols.** In this work, we focus on MPC-based secure inference protocols for neural networks which started with the seminal work of SecureML (Mohassel & Zhang, 2017). SecureML considers the two-party setting that only involves the service provider and the client, and after many follow-up works in this setting (Mohassel & Zhang, 2017; Juvekar et al., 2018; Liu et al., 2017; Mishra et al., 2020; Rathee et al., 2020; 2021; Zhang et al., 2021; Huang et al., 2022; Balla & Koushanfar, 2023; Hao et al., 2022; Hou et al., 2023; Lu et al., 2025; Pang et al., 2024), the performance has improved by orders of magnitude. Despite these improvements, 2PC still poses very large overheads. Thus, subsequent works have considered other settings that introduce an additional helper party such as 3PC with honest majority (Wagh et al., 2019; Kumar et al., 2020; Riazi et al.,

2018; Mohassel & Rindal, 2018; Wagh et al., 2021; Dong et al., 2023) and 2PC with trusted dealer (2PC-Dealer) (Knott et al., 2020; Gupta et al., 2022; Jawalkar et al., 2024; Gupta et al., 2024). Other works have accelerated secure inference protocols by leveraging GPU acceleration (Knott et al., 2020; Tan et al., 2021; Watson et al., 2022; Jawalkar et al., 2024; Gupta et al., 2024).

Recent work (Hao et al., 2022; Hou et al., 2023; Lu et al., 2025; Pang et al., 2024; Dong et al., 2023; Wang et al., 2022; Gupta et al., 2024) in all these settings have focused on secure transformer inference since they represent the majority of the AI workload today. Our work is orthogonal to these protocols and can be used to accelerate secure inference with any of them (Appendix H).

**MPC-friendly Approximations.** Several works (Li et al., 2023b; Mohassel & Zhang, 2017; Gilad-Bachrach et al., 2016; Ghodsi et al., 2020a; Chou et al., 2018; Chen et al., 2022; Mishra et al., 2020; Luo et al., 2024; Jha et al., 2021; Peng et al., 2023; Cho et al., 2022b;a; Lou et al., 2020; Kundu et al., 2023; Zhang et al., 2023) have proposed approximate implementations for non-linear activations like softmax and GeLU to make them more MPC-friendly. These approximations typically introduce a large drop in model performance. MPCFormer (Li et al., 2023b) proposed a two-stage distillation process to bridge this gap. Majority of these works (Mishra et al., 2020; Jha et al., 2021; Ghodsi et al., 2020b; Peng et al., 2023; Cho et al., 2022b;a; Kundu et al., 2023; Lou et al., 2020; Zhang et al., 2023) also use Neural Architecture Search (NAS) to employ multiple approximations within the same network depending on the precision level required. We expand on how MARILL is different from MPC-friendly approximations in Appendix D.1.

## 3 THREAT MODEL AND SETTING

In the secure inference setting, there is a user with a private input and a logical server with a private model. The model architecture is assumed to be public, the service provider only wants to hide the private model weights, and the user wants to hide its private input.

**Traditional threat model.** We focus on the threat model commonly considered by prior works on secure transformer inference: a semi-honest (or passive) adversary that compromises either the user or the logical server. In settings where the logical server is implemented via a set of MPC participants, the adversary compromises an unknown subset of these participants (see Appendix A). MARILL is not limited to a semi-honest adversary and we discuss malicious security in Appendix F.

**Open-source pre-trained model setting.** MARILL applies to the emerging open-source pre-trained model setting where the service provider fine-tunes a (public) open-source model with its private data to produce an inference model. Unlike prior works, MARILL produces models where not all weights must be kept private from the user. We now explain why MARILL's models uphold the principles of secure inference despite using a mix of public and private weights:

- **Public-private architecture does not leak private data**: MARILL statically determines which weights should be kept public or private through simple configurations (layers frozen, LoRA rank), and this is done independently of the private dataset. The public weights in a MARILL-generated model are exactly the open-source model weights frozen during fine-tuning, and thus, making them public does not reveal any extra information about the private weights or the private dataset.
- **Public parts of the model evaluated outside MPC do not leak additional data**: since MARILL is black-box in the underlying MPC protocol, all that remains to be proven is that the public parts MARILL evaluates outside MPC do not leak additional data about the private inputs. The proof is straightforward (see Appendix E) and we show that evaluating just the private part of the model within MPC is equivalent to evaluating the whole model within MPC, i.e., client only learns the output tokens and the server learns nothing. Given just the output tokens, what can be learned about the private weights is an orthogonal question (see Appendix D.3 for discussion on recent work).

## 4 PERFORMANCE CHARACTERISTICS OF SECURE INFERENCE

Secure inference relies on secure multi-party computation (MPC) (Goldreich et al., 1987; Yao, 1986), a cryptographic primitive that allows mutually distrusting parties to compute any function on their private inputs without revealing anything beyond the function output. Prior secure inference works, specifically, have considered three MPC settings (Appendix A), each making different assumptions

about the participants. In this section, we highlight the unique cost profile of MPC in these settings and discuss how it motivates the design of our techniques in § 5.

**Interaction costs.** Unlike plaintext computation, most operations within MPC require interaction among the MPC participants. This imposes two additional performance overheads in addition to computation size, namely, *communication size* and *rounds of communication*. For most MPC protocols, this cost of interaction ends up being the bottleneck and it is the primary reason why MPC is orders of magnitude slower than plaintext computation.

**Multiplications with public weights come for free.** Since MPC operates natively over integers, recent secure inference works use fixed-point representation to emulate real-number arithmetic. Additionally, prior works maintain the invariant that the intermediate state after every network layer is arithmetically secret-shared (ASS) among MPC participants. This approach minimizes the cost of arithmetic operations, such as integer multiplications and additions, which dominate ML workloads. In an ASS scheme, a secret value $x$ is split among $n$ MPC participants such that (i) each party $\mathcal{P}_i$ receives a share $x_i$ and any set of $n - 1$ shares reveals nothing about $x$, and (ii) the sum of all shares reconstructs the secret $x = x_1 + \ldots + x_n$. The linear nature of this reconstruction function allows secret-shared values to be added locally (without interaction) by simply adding the corresponding secret shares, making additions within MPC relatively so inexpensive that they are considered "free". Similarly, any affine operation with public coefficients on secret-shared values, such as a matrix multiplication with public weights, also becomes free. In § 5.2, we show how low-rank adaptations can leverage this property to reduce the number of multiplications between secret-shared values.

**Non-arithmetic operations are the bottleneck in the most efficient MPC settings.** Non-arithmetic operations are used to implement comparisons in maxpool, activation functions such as ReLU and GeLU, exponentiation and division in softmax, as well as the truncation operations in fixed-point multiplications. We analyzed state-of-the-art secure inference frameworks (§ 6.1) in the most efficient MPC settings, namely, 3PC and 2PC-Dealer (Appendix A), and found that non-arithmetic operations account for over 88% of the runtime and communication during secure inference with a sequence length of 2048. This is in stark contrast to plaintext computation where non-arithmetic operations have a minimal contribution to the total FLOPs and the inference latency. Guided by this insight, we proposed head-merging in § 5.3, a technique that preserves the FLOPs and still yields significant performance improvements.

**A mix of public and private weights typically does not speedup secure inference.** Since multiplications with public weights come for free, one would expect significant improvements to secure inference if most of the weights were public. However, to preserve the standard guarantees of the MPC, an intermediate state that depends on both the private input and any private weight must not be revealed to any party. Consequently, once the computation involves a single private weight, all subsequent non-arithmetic operations need to be performed within MPC, which as we just discussed are the bottleneck in the most efficient MPC settings for secure inference. This restriction motivated the design of layer-freezing in § 5.1, which separates the public and private weights across layers such that the non-arithmetic operations in public layers are performed outside MPC.

## 5 TECHNIQUES

In this section, we describe our techniques that minimize the need for expensive operations within MPC. We start with layer-freezing (§ 5.1) that reduces the number of layers evaluated within MPC. Next, we discuss LoRA (§ 5.2) and head-merging (§ 5.3) that minimize arithmetic and non-arithmetic operations, respectively, in the private layers. Distillation details are deferred to Appendix G.

### 5.1 LAYER FREEZING

We start with the observation that when an open-source model is fine-tuned on a private dataset, only the fine-tuned weights need to be kept private during inference. To leverage this insight, consider using a technique from prior work that only fine-tunes a fraction of model weights (Gandhi et al., 2023). However, as explained in § 4, these techniques typically do not significantly speed up inference. This is because they update weights throughout the network, including near the input, which means that almost all non-arithmetic operations – typically the bottleneck – must be performed within MPC.

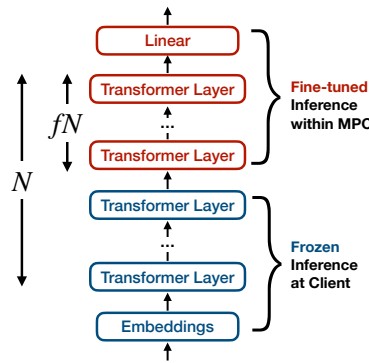

(a) Layer Freezing with fraction $f$ layers fine-tuned

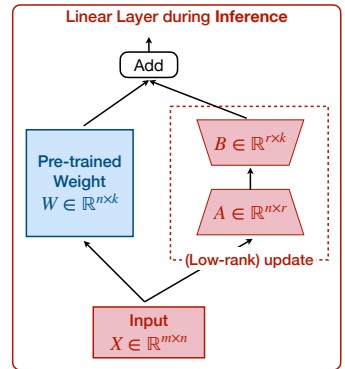

(b) Linear layer during inference with LoRA

Figure 2: MARILL's techniques that leverage public weights (marked in blue).

To this end, our solution (Fig. 2a) effectively leverages public weights by deferring fine-tuning to only the final layers of the transformer, thereby also deferring MPC to these final layers. During inference, the client receives the weights for the bottom layers (identical to the open-source pre-trained model) from the server, computes the output of these layers locally, and then engages in MPC with the server for the top layers. Consequently, if only a fraction $f$ of the layers are fine-tuned, all MPC overheads are reduced by a factor of $\frac{1}{f} \times$ (Table 2). Although delegating the computation of the bottom layers to the client might seem like a limitation, this approach actually *reduces client overheads* by the same factor, since the MPC overhead on the client in secure inference protocols is orders of magnitude higher than the overhead of plaintext inference[2].

## 5.2 LoRA ADAPTATION

In § 4, we discussed how multiplication with public weights is free during secure inference. Here, we demonstrate how LoRA (Hu et al., 2022), a technique developed for parameter-efficient fine-tuning, can be repurposed to minimize integer multiplications during inference. These operations account for up to $95\%$ of the runtime in the state-of-the-art 2PC work Bumblebee (Lu et al., 2025). Beyond the 2PC setting, we found that multiplications also dominate the decoding (see Appendix B) runtime in 3PC and 2PC-Dealer settings, which are otherwise bottlenecked by non-arithmetic operations (§ 4). This occurs because the linear layers during decoding perform matrix-vector multiplications instead of matrix multiplications, making key matrix-multiplication optimizations from Mohassel & Zhang (2017) no longer applicable.

A LoRA adapter on a weight matrix $W \in \mathbb{R}^{n \times k}$ is a product of two low-rank matrices $A \in \mathbb{R}^{n \times r}$ and $B \in \mathbb{R}^{r \times k}$, where $r \ll \min(n, k)$. During fine-tuning, only the low-rank matrices are updated, and at inference time, $A \times B$ is merged into the pre-trained weight $W$ to minimize inference overhead. This approach updates all the model weights and we do not get any benefit from the public pre-trained weights. In our solution, we crucially *do not* merge the product $A \times B$ with the pre-trained model weights and keep the matrices separate as shown in Fig. 2b. To see why this reduces multiplications, consider the evaluation of a LoRA-adapted linear layer: for input $X \in \mathbb{R}^{m \times n}$, the evaluation function can be written as $X \times (W + A \times B)$. Naïvely, the complexity of this expression is $O(mnk)$. However within MPC, the product $X \times W$ comes for free (§ 4). To evaluate the remaining expression $X \times A \times B$, instead of computing $A \times B$ first, we can first evaluate $X \times A$ and then multiply it with $B$. This reduces the overall complexity to $O(mr(n + k))$; for $n = k = 3200$ and $r = 64$, this idea reduces the number of multiplications by $25\times$.

## 5.3 HEAD MERGING

The most efficient secure inference works (Dong et al., 2023; Knott et al., 2020; Gupta et al., 2024) operate in the 3PC and the 2PC-Dealer settings (Appendix A). In these settings, non-arithmetic operations are the bottleneck. Among these operations, those in the self-attention module are

---

[2]The overhead on MPC participants, including the client, is nearly identical in all secure inference protocols, and even the state-of-the-art protocol has a $73\times$ overhead over plaintext inference (Gupta et al., 2024).

of particular interest because: (i) the self-attention mechanism is the only component that scales quadratically with sequence length $b$, (ii) the state-of-the-art works in both 3PC (Dong et al., 2023) and 2PC-Dealer (Gupta et al., 2024; Knott et al., 2020) settings exhibit a super linear blowup in runtime when $b \geq 1024$, highlighting that self-attention is indeed the bottleneck for large $b$, and (iii) applications such as chatbots and copilots which have real-time requirements require a large sequence length. Thus, we focus on minimizing the non-arithmetic operations in the self-attention module.

**Reducing number of heads.** Only the scaled dot-product attention (SDPA) module within the self-attention mechanism has non-arithmetic operations that scale quadratically with $b$. These operations are softmax and truncations (from fixed-point multiplications), and the complexity for both is $O(b^2 h)$, where $h$ is the #heads. Hence, we seek to reduce $h$ by a factor $m$ to reduce both operations proportionally. The standard technique for minimizing heads is head-pruning (Michel et al., 2019), which analyzes the importance of each head over the training dataset, and prunes the insignificant heads. This achieves our goal, but we have to prune $75\%$ of the heads (and their parameters) for $m = 4$, and this results in a large accuracy loss (§ 6.4).

**Preserving the pre-trained parameters.** To this end, we observe that unlike plaintext inference, FLOPs do not dictate the secure inference cost (§ 4) and it is possible to achieve similar speedups as head-pruning despite preserving all the parameters (§ 6.4). This is also evident in

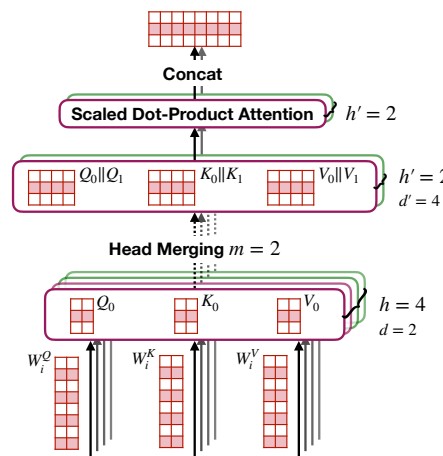

Figure 3: Head merging ($m = 2$) example for seq-len $b = 3$, #heads $h = 4$, and head-dim $d = 2$. After merging, $h$ reduces to $h' = 2$ and $d$ increases to $d' = 4$. The red matrices represent that head-merging is only performed in private layers.

the complexity of non-arithmetic operations in self-attention, which are independent of the head-dimension $d$. Thus, we propose a technique called head-merging that reduces the number of heads $h$ by $m\times$, while simultaneously increasing the head dimension $d$ proportionally, thereby preserving all parameters from the pre-trained model. Specifically, $h$ heads are divided into groups of $m$, and the QKV matrices for heads within the same group are concatenated as shown in Fig. 3. Concretely, given matrices $\{Q_i, K_i, V_i\}_{i \in [h]}$ of dimension $\mathbb{R}^{b \times d}$, the head attention outputs $\{\mathsf{head}_j\}_{j \in [h/m]}$ after merging are as follows: $\mathsf{head}_j = \mathsf{softmax}\left(\frac{\sum_{\ell=jm}^{(j+1)m} Q_\ell K_\ell^T}{\sqrt{md}}\right) \cdot \left(V_{jm} \| \cdots \| V_{(j+1)m}\right) \in \mathbb{R}^{b \times md}$.

**Merging similar heads.** In the expression above, adjacent heads are grouped such that heads $jm$ to $(j + 1)m$ belong to group $j$. This strategy does not consider the similarity among heads, resulting in minimal accuracy improvement over head-pruning (§ 6.4). To group heads based on similarity, we follow the strategy from (Bian et al., 2021) that computes the pairwise Jensen-Shannon distance between all heads within the same layer. Once we have the pairwise distances, we perform K-Medoid clustering (Kaufman, 1990) to organize heads into $h/m$ groups. Finally, to get groups of the same size, we redistribute heads based on a linear sum assignment that minimizes the sum of distances from the medoid within each group. We found that merging similar heads using this method performs significantly better, leading to up to $8\%$ gain in accuracy § 6.4.

# 6 EVALUATION

In this section, we first evaluate the secure inference cost (§ 6.1) of MARILL-generated models and their ability to preserve ML performance (§ 6.2). Next, we perform the same analysis for prior MPC-friendly approximations integrated with MARILL (§ 6.3). Finally, we do an ablation study in § 6.4 that considers alternative designs for MARILL's techniques.

**Secure Inference Setup.** We perform secure inference experiments on the state-of-the-art (SOTA) secure inference frameworks: SPU (Ma et al., 2023), which supports SOTA protocols for 2PC (Lu et al., 2025) and 3PC (Dong et al., 2023), and Crypten (Knott et al., 2020) which is a popular framework in the 2PC-Dealer setting. Additionally, we evaluate SIGMA (Gupta et al., 2024), the

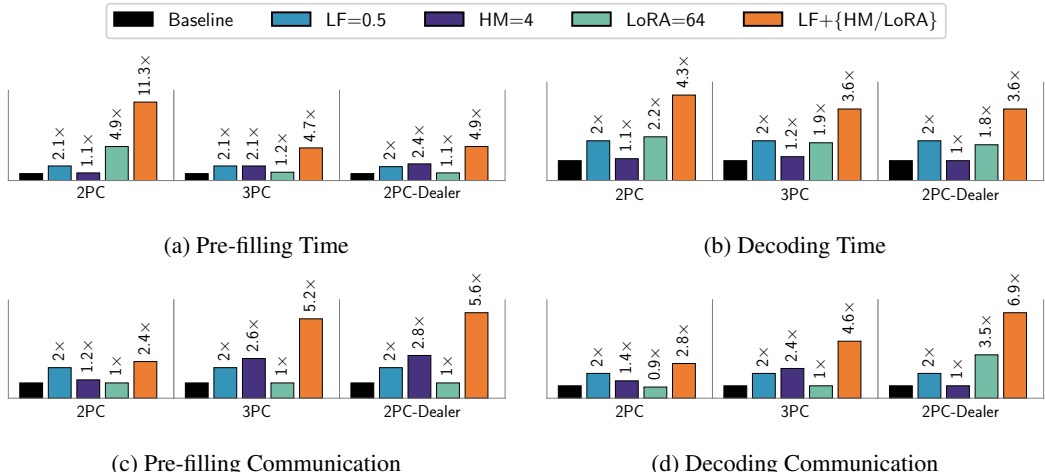

Figure 4: Secure inference performance of MARILL vs standard fine-tuning for `openllama-3b-v2` in the LAN setting. The sequence length is $b = 64$ for 2PC and $b = 2048$ for 3PC and 2PC-Dealer. Bar labels show improvement factors over the baseline. The final bar in each plot represents the combination of layer-freezing with head-merging or LoRA, whichever performs better independently.

SOTA 2PC-Dealer protocol and defer its results to Appendix J. The experiments were run on two or three machines (depending on the MPC setting) in two network settings: a LAN connection (16 Gbps bandwidth, 0.1 ms latency) and a WAN connection (400 Mbps bandwidth, 40 ms latency). Each machine was equipped with an Intel Xeon Platinum 8173M Processor with 16 vCPUs, 128 GB RAM, and a $V100$ GPU with 16 GB memory. Since the 2PC-Dealer framework (Knott et al., 2020) supports GPU acceleration, we ran it on the V100. Experiments on other MPC frameworks were run on CPU. All experiments were multi-threaded. All reported numbers consider end-to-end costs.

**Models and Datasets.** We consider three privacy-sensitive tasks for LLMs: chatbot, coding, and machine translation. For the chatbot task, we fine-tune `open-llama3b-v2` on the ShareGPT dataset and evaluate it on the MTBench dataset, following Zheng et al. (2023); Li et al. (2023a). OpenLLaMA is a popular open-source model that replicates the LLaMA model (Geng & Liu, 2023; Touvron et al., 2023). For the coding task, we fine-tune `deepseek-coder-1.3b-base` on the MagiCoder dataset (Wei et al., 2023) and evaluate it on the HumanEval benchmark (Chen et al., 2021). For the machine translation task, we fine-tune `open-llama3b-v2` on the ParroT dataset (Jiao et al., 2023) and evaluate it on the WMT22 (De⇒En) benchmark (Kocmi et al., 2022).

**Fine-Tuning Hyperparameters.** We set the fine-tuning hyperparameters according to the papers that curated the corresponding fine-tuning dataset: Zheng et al. (2023) for MTBench, Wei et al. (2023) for HumanEval, and Jiao et al. (2023) for WMT22. We only vary the batch size and number of training epochs to better suit some techniques. For instance, we observed that LoRA favors a smaller batch size in our setting. We include the detailed hyperparameters in Appendix I.

## 6.1 SECURE INFERENCE PERFORMANCE

In this section, we compare the secure inference performance of MARILL-generated models vs the baseline – a fully fine-tuned model. We focus on LAN performance in this section and defer the discussion on WAN performance to Appendix K. Fig. 4 summarizes the LAN results for `openllama-3b-v2` as the pre-trained model. We first analyze the improvements from head-merging (§ 5.3) and LoRA (§ 5.2) in the three MPC settings from prior work, and then discuss layer-freezing (§ 5.1) improvements.

**2PC**: LoRA improves the pre-filling runtime by $4.9\times$ (Fig. 4a) as 92% of the 2PC runtime is spent in performing multiplications for `openllama-3b-v2` inference. Decoding runtime is improved by $2.2\times$, which is less pronounced because the 2PC framework (Lu et al., 2025) does not amortize well over the smaller decoding computation. In terms of communication, non-arithmetic operations are the bottleneck in 2PC, accounting for 72.5% of the total communication. Still, we do not see a large

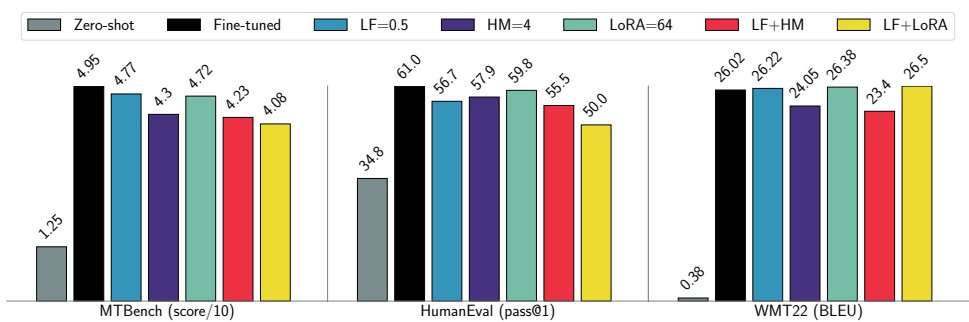

Figure 5: MARILL vs (fully) fine-tuned and zero-shot baselines.

improvement with head merging (Figures 4c & 4d) because it is designed for large sequence lengths and we could only run 2PC on small sequence lengths (64) due to its large memory requirements.

**3PC and 2PC-Dealer**: Since non-arithmetic operations in the self-attention module become the bottleneck in these settings at large sequence lengths (§ 5.3), head-merging leads to runtime and communication improvements of $2.1 - 2.4\times$ (Fig. 4a) and $2.6 - 2.8\times$ (Fig. 4c), respectively, in the pre-filling stage. During decoding, integer multiplications are the runtime bottleneck instead (§ 5.2), and hence, LoRA helps in this stage and we get $1.8 - 1.9\times$ (Fig. 4b) decoding runtime improvement. In terms of decoding communication (Fig. 4d), 3PC exhibits a similar improvement as in pre-filling. The communication improvement from LoRA for 2PC-Dealer is an implementation artefact[3].

**Layer Freezing (§ 5.1)**: We fine-tune half of the 26 transformer layers in `openllama-3b-v2`. This leads to around $2\times$ improvement across settings, metrics, inference stages, and in combination with both techniques. In some cases, layer freezing leads to a greater than $2\times$ improvement due to the omission of the embedding layer within MPC in addition to half of the transformer layers. In general, we show in Table 2 that layer freezing leads to $\frac{1}{f}\times$ improvement in all metrics for a wide range of $f$ values. Overall, including WAN results (Appendix K), MARILL leads to $2.2 - 11.3\times$ better runtime and $2.4 - 6.9\times$ better communication across secure inference scenarios.

## 6.2 ML PERFORMANCE

Fig. 5 summarizes the ML performance of MARILL, the pre-trained model and the fully fine-tuned model on our three benchmarks. First, we note that full fine-tuning significantly improves the performance of the pre-trained model across all three tasks. MARILL's layer-freezing (LF=0.5) is also effective on all three tasks, preserving $93 - 100\%$ of the full fine-tuning performance (see Appendix L for ablation on number of layers frozen). On WMT and HumanEval benchmark, head-merging (HM=4) preserves $92 - 95\%$ performance, while on MTBench, it achieves $87\%$ performance. The combination of layer-freezing and head-merging works well, incurring an additional loss of at most $4\%$ compared to head-merging alone. For scenarios requiring higher accuracy, the HM=2 configuration offers significantly improved accuracy while still outperforming the baseline (Table 1b). LoRA preserves over $95\%$ performance on all benchmarks. While combining LoRA with layer freezing sometimes leads to a big drop in performance (MTBench and HumanEval), we note that using LoRA alone provides significant speed-ups, ranging from $2.2\times$ to $4.9\times$. Overall, we observe that MARILL's techniques typically preserve over $90\%$ of the fully fine-tuned performance.

## 6.3 INTEGRATION OF PRIOR MPC-FRIENDLY APPROXIMATIONS WITH MARILL

In this section, we analyze the performance of MARILL when combined with prior MPC-friendly approximations, namely, Quad (Li et al., 2023b) and ReLU (Chen et al., 2022; Zeng et al., 2023) as GeLU/SiLU approximations, and 2Quad (Li et al., 2023b), L2Quad (Zhang et al., 2023) and 2ReLU (Mohassel & Zhang, 2017) as softmax approximation. First, we analyzed the ML performance of each approximation independently and found that the quadratic approximations from recent works led to a catastrophic loss on our benchmarks. Specifically, on the HumanEval benchmark, Quad only

---

[3]We had to employ matrix decomposition on all linear layers in the 2PC-Dealer setting to fit secure inference of (fully) fine-tuned LLaMA-3B on the V100 GPU.

Table 1: HumanEval pass@1 performance of various techniques. The time and comm. improvements are averaged over the prefilling stage in the 3PC and 2PC-Dealer settings on a LAN network.

(a) 2ReLU approximation for softmax combined with MARILL (LF=0.5, HM=4)

| | pass@1 | Improvement Time | Comm. |
|---|---|---|---|
| HM=4 | 57.9 | 2.25× | 2.7× |
| 2ReLU + HM | 54.9 | 3.25× | 4.25× |
| LF=0.5 + HM=4 | 55.5 | 4.8× | 5.4× |
| 2ReLU + LF + HM | 56.7 | 6.9× | 8.5× |

(b) Adjacent/similar head-merging vs head-pruning (HP). Parameter denotes the head reduction factor.

| | pass@1 | Improvement Time | Comm. |
|---|---|---|---|
| HP=4 | 49.4 | 2.45× | 2.75× |
| HP=2 | 56.7 | 1.7× | 1.8× |
| HM=4 (adj.) | 50.0 | 2.25× | 2.7× |
| HM=4 (sim.) | 57.9 | 2.25× | 2.7× |
| HM=2 (sim.) | 60.4 | 1.55× | 1.8× |

achieves 31.7% accuracy compared to 61% of the baseline, and the fine-tuning diverges for L2Quad and 2Quad, resulting in 0% accuracy. In contrast, ReLU-based approximations work very well, with ReLU achieving the same accuracy as the baseline, and 2ReLU achieving 59.8% accuracy. Out of the two, only 2ReLU leads to significant efficiency improvements, with ReLU only improving the secure inference cost by at most 10%. Thus, we only evaluate the combination of 2ReLU with MARILL.

Table 1a summarizes the accuracy results on the HumanEval benchmark and the corresponding secure inference improvements. For the latter results, we focus on the 3PC and 2PC-Dealer settings because all prior approximations target non-arithmetic operations that are the bottleneck in these settings. Our experiments show that 2ReLU works well with MARILL, incurring at most 3% further accuracy loss on top of MARILL. In exchange, 2ReLU improves MARILL's time and communication by $1.4 - 1.6\times$. For reference, 2ReLU independently results in $1.95 - 2.15\times$ improvement over the baseline. Overall, we get $6.9 - 8.5\times$ improvement in runtime and communication compared to the baseline, while still preserving over 90% of the baseline ML performance.

### 6.4 ABLATION STUDY

**Layer-freezing vs layer-pruning.** In layer-freezing, we froze the bottom layers of the transformer to move some layers outside of MPC. An alternative strategy to minimize layers within MPC is to simply prune some layers. We experimented with layer-pruning on the HumanEval benchmark and evaluated the best-performing strategy from Sajjad et al. (2020), namely, top-layer pruning. For half of the layers pruned, we found that the accuracy drops from 61% for the baseline to just 49.4% post layer-pruning. In contrast, layer-freezing achieved an accuracy of 56.7%, a 12% increase in relative performance, highlighting the importance of preserving the pre-trained weights of the pruned layers.

**Head-merging vs head-pruning.** We compared head-pruning (Michel et al., 2019) and head-merging § 5.3 on HumanEval, configuring head-pruning to prune an equal number of heads from each layer to avoid additional leakage about the private dataset. Table 1b summarizes the results for both techniques when the heads are reduced by $2\times$ and $4\times$. First, we note that head-merging achieves similar efficiency improvements to head-pruning for both head reduction factors, with head-pruning being at most 10% faster and 2% more communication efficient. ML performance of head-merging, on the other hand, is much better since it preserves all the head parameters. In particular, head-merging has up to 8% better accuracy than head-pruning, and HM= 4 even outperforms HP=2 in both ML and secure inference performance. Note that these improvements only apply to similar head-merging, not adjacent head-merging, which naïvely combines adjacent heads. These results demonstrate the significance of preserving head parameters as well as merging heads based on similarity.

## 7 CONCLUSION

In this work, we designed a framework MARILL, that leverages open-sourced LLMs and introduces high-level architectural changes through fine-tuning to minimize MPC usage during secure inference. We demonstrated that MARILL is effective in minimizing secure inference costs across MPC settings in exchange for a reasonable accuracy tradeoff. In particular, MARILL-generated models are $2.2 - 11.3\times$ more efficient for secure inference compared to a standard fine-tuned model, and they typically preserve over 90% relative performance across multiple challenging LLM tasks.

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

## A  MPC Settings

- **2-party computation (2PC)**: this setting assumes two MPC participants who do not trust each other, and thus, it is the most natural setting for secure inference.

- **Honest-majority 3-party computation (3PC)**: this setting has an additional helper party that also participates in MPC, and the adversary can corrupt at most any one of the three parties. Prior works considered this setting because having this helper party improves the MPC performance by orders of magnitude.

- **2PC with trusted dealer (2PC-Dealer)**: in this setting, there is an additional trusted dealer that is only responsible for distributing *input-independent* correlated randomness to the computing parties in a pre-processing phase. The parties can then use this randomness to accelerate 2PC on their private inputs.

## B  LLM Inference Stages - Prefilling and Decoding

In this section, we briefly describe the two stages in LLM inference. Firstly, users provide a prompt in natural language to the system. The system then uses tokenizers to map the natural language into a vector $x_1, ... x_n$ through a process called tokenization (Sennrich et al., 2015). Then the system performs the main inference process using LLMs. The inference process consists of two phases - the pre-filling phase and the decoding phase. Formally, the pre-filling phase computes probablity of the first token conditioned on the previous $n$ tokens $P(x_{n+1}|x_1, ... x_n)$ (Sheng et al., 2024). It then samples from the distribution and predicts the first token $x_{n+1}$. The decoding phase iteratively computes the next token based on the same logic. For instance, the first step in the decoding computes $P(x_{n+2}|x_1, ... x_{n+1})$ and samples to obtain $x_{n+2}$. The decoding phase terminate when the new token is an ending token, often referred to as the "end-of-sentence" token (EOS). Interestingly, the left-to-right decoding nature has made the computation characteristics different (Kwon et al., 2023; Yu et al., 2022; Sheng et al., 2024) in these two stages. Thus, we distinguish between the two phases when evaluating our techniques in this work.

## C  Marill Does Not Accelerate Plaintext Inference

Marill introduces changes during fine-tuning that minimize MPC usage during secure inference. In this section, we argue that these changes are specifically tailored to reduce secure inference costs and do not accelerate plaintext inference:

- **Layer freezing** (§ 5.1): although layer freezing restricts the private weights to just the final layers, plaintext inference still has to evaluate all the layers, resulting in no performance improvement. Secure inference also evaluates all the layers, however, the expensive MPC overhead is only paid for the layers with private weights.

- **LoRA** (§ 5.2): LoRA introduces low-rank matrices $A$ and $B$ to every weight matrix $W$ of the model such that the linear layer computation on input $X$ becomes $(W + A \times B) \times X$, as opposed to $W \times X$ without LoRA. Thus, running plaintext inference with LoRA actually increases overhead. Again, secure inference also evaluates $(W + A \times B) \times X$ for all linear layers, however, the expensive MPC overhead is only paid for computing $A \times B \times X$ since multiplication with public weights comes for free in MPC (§ 4).

- **Head merging** (§ 5.3): head-merging reduces the number of heads $h$ but also proportionally increases the head-dimension $d$ such that $c = h * d$ remains the same. Plaintext inference cost is dictated by the FLOP count, which for self-attention is $O((b^2 + bc) \cdot c)$ where $b$ is the sequence length. Note that head merging does not change the FLOP count, and thus the plaintext inference cost remains the same. Secure inference benefits from head merging because it reduces non-arithmetic operations that have a negligible contribution to FLOP count but are the bottleneck in MPC due to its unique cost profile (§ 4).

## D  RELATED WORK

### D.1  MARILL VS MPC-FRIENDLY APPROXIMATIONS.

MARILL is complementary to MPC-friendly approximations as it makes high-level changes to the architecture, as opposed to the underlying operations. Additionally, MARILL differs from these works in two key aspects: (i) while these works output models where all weights are private, MARILL produces models that have a mix of public and private weights, and (ii) the model architecture in NAS-based works depends on the private training data and leaks additional information, whereas MARILL is statically configured independent of the training data. We show in § 6.3 that these approximations can be combined with MARILL to yield further performance improvements. We do not include NAS-based approximations here because they leak information beyond the standard secure inference guarantees, which conflicts with the strict security requirements we aim to preserve.

### D.2  LAYER-FREEZING AND LoRA IN TEE-BASED SECURE ML

Both MARILL and prior work on TEE-based secure ML (Zhang et al., 2024; Huang et al., 2024) divide the model weights into public and private components to optimize computation (either for secure training or inference), and the specific techniques of interest are LoRA and layer freezing. In this section, we highlight the key differences in how these techniques are adapted in each setting.

- **LoRA**: In TEE-based works, LoRA is used to shift computations involving public weights outside the TEE during inference, provided these computations are preprocessed securely within the TEE during an equally expensive inference-specific preprocessing phase (Tramèr & Boneh, 2019). In constrast, MARILL performs all LoRA-related computations (on both public and private weights) within the MPC environment, leveraging the observation that arithmetic operations on public weights have the same cost as the corresponding plaintext operation (§ 4). As a result, MARILL can accelerate secure inference with LoRA without requiring a preprocessing phase.

- **Layer-Freezing**: Prior work on TEE-based secure inference (Zhang et al., 2024; Mo et al., 2020; Elgamal & Nahrstedt, 2020; Hou et al., 2022; Shen et al., 2022; Sun et al., 2023) has adopted alternating public and private layers. While effective for TEE-based secure inference (where arithmetic operations are the bottleneck), this approach offers limited benefits in MPC-based settings (e.g., 2PC-Dealer and 3PC) since the bottleneck in these settings lies in non-arithmetic operations that must still occur within MPC (§ 4). Instead, MARILL introduces a strict separation between public and private components to optimize secure inference. Finally, while the recent work on TEE-based secure federated learning (Huang et al., 2024) employs (split) layer-freezing like MARILL, and this idea can be extended to TEE-based secure inference, we employ this technique in the MPC-based secure inference context for the first time and provide new and comprehensive results on its accuracy (in the traditional non-federated setting) and MPC performance (Table 2).

### D.3  MODEL EXTRACTION.

So far, the state-of-the-art model extraction attacks (Canales-Martínez et al., 2024; Carlini et al., 2024; Chen et al., 2024; Foerster et al., 2024) that can benefit from some weights being public have major limitations: they (i) make assumptions that do not apply to our setting, (ii) only scale to orders of magnitude fewer parameters than the private weights in any configuration of MARILL we evaluate, and (iii) require a very large number of queries which are infeasible to perform given the high cost of secure inference.

## E  SECURITY PROOF

In this section, we prove that secure inference on MARILL-generated models satisfies the standard guarantee of secure inference, i.e., the user only learns the output tokens and the server learns nothing from a secure inference execution. Since MARILL is black-box in the underlying secure inference protocol and the model architecture itself does not leak private data, all we need to prove is that evaluating public parts of MARILL-generated model outside MPC does not reveal additional information. We prove the same by showing that evaluating just the private part of the model within MPC is equivalent in terms of security to evaluating the whole model within MPC.

First, we describe the ideal functionality that models the black-box functionality provided by a secure inference protocol. Then, we discuss how we model the public-private architecture of MARILL's models in the proof. Finally, we explain the high-level intuition of our proof, and conclude with a formal proof.

---

**Secure Inference Ideal Functionality $\mathcal{F}_A$**

This functionality is parameterized by the model architecture $A$.

- **Client Prompt**: Receive prompt $p$ for $A$ from client $\mathcal{C}$, and store $p$ internally.

- **Server Weights**: Receive model weights $W$ for $A$ from server $\mathcal{S}$, store $W$ internally.

- **Pre-filling**: Perform pre-filling on $p$ to get state $\mathsf{st} \leftarrow A.\mathsf{prefill}(W, p)$. Set $y_{\mathsf{prev}} \leftarrow \bot$.

- **Decoding**: If $y_{\mathsf{prev}} \neq \bot$, receive token $x$ from the client $\mathcal{C}$. If $x = y_{\mathsf{prev}}$, update the state $\mathsf{st} \leftarrow A.\mathsf{update}(\mathsf{st}, x)$; else abort. Perform a decoding step on $\mathsf{st}$ to get an output token $y \leftarrow A.\mathsf{decode}(\mathsf{st})$, update $y_{\mathsf{prev}} \leftarrow y$, and send $y$ to the client $\mathcal{C}$.

---

Figure 6: Ideal functionality for secure inference

**Ideal functionality of secure inference.** The secure functionality provided by a secure inference protocol can be described using a (simplified) ideal functionality $\mathcal{F}_A$ (Fig. 6) that is parameterized by a model architecture $A$. Note that $\mathcal{F}_A$ does not leak any information to the server, and the client learns nothing beyond the output tokens. The ideal functionality allows the client to choose the latest token $x$, but the ideal functionality makes sure that this token must match the previously generated token $y_{\mathsf{prev}}$. It is important to note this ideal functionality is simplified for exposition and there are some additional considerations when it is realized with a secure inference protocol in practice:

- Secure inference protocols typically emulate real-number arithmetic with fixed-point arithmetic which incurs numerical errors. Thus, the functionality needs to be modified to faithfully perform each operation according to the fixed-point schema used by the specific secure inference protocol.

- Many of the secure inference protocols (Demmler et al., 2015; Knott et al., 2020; Mohassel & Zhang, 2017; Wagh et al., 2019; Lu et al., 2025; Dong et al., 2023; Mishra et al., 2020; Tan et al., 2021; Wagh et al., 2021; Huang et al., 2022) we cite employ probabilistic truncation to boost efficiency, and it was shown in Li et al. (2023c) that these protocols can not be proved secure w.r.t. any ideal functionality. Thus, our proof only applies to protocols that do not have this limitation.

**Modelling the public-private architecture of MARILL's models.** MARILL introduces three techniques that make the following modifications to the model architecture:

- Layer freezing: it splits the model into public and private layers, where the bottom layers are all public and the top layers are all private.

- LoRA: it makes the majority of weights in the private layers public.

- Head-merging: it changes the number of heads in the private layers.

It is straightforward to argue that LoRA and head merging satisfy the standard secure inference guarantee because they only impact the private layers which are entirely run within MPC. Even for LoRA, the public weights just make the MPC much more efficient (§ 4) but the computation is still run entirely within MPC. The changes they make can simply be seen as running a different architecture within MPC, and MPC ensures that only the output of the private layers (i.e., the output tokens) is revealed. Thus, we only need to prove that splitting the model into public and private layers is secure because this actually moves operations outside MPC. To this end, we model the public-private architecture as follows: $M_{\mathsf{pb}}$ denotes the public layers run outside MPC, and $M_{\mathsf{pr}}$ denotes the private layers run within MPC. From our layer-freezing strategy, $M = M_{\mathsf{pb}} \| M_{\mathsf{pr}}$ denotes the complete inference architecture, which is basically a concatenation of the public layers with the private layers. Now, we look at the security proof which proves that only evaluating $M_{\mathsf{pr}}$ within MPC has the same security as running $M$ entirely within MPC.

**High-level proof strategy.** MARILL's secure inference protocol (Fig. 7) evaluates $M_{\mathsf{pr}}$ within MPC by making black-box calls to a prior secure inference framework. Our proof shows that MARILL's

---

**MARILL's Secure Inference Protocol in the $\mathcal{F}$-hybrid model**

Let $M$, $M_{\mathsf{pb}}$, and $M_{\mathsf{pr}}$ denote the model architecture components as defined in Appendix E. Let $W_{\mathsf{pb}}$ and $W_{\mathsf{pr}}$ denote the corresponding weights for these parts. Client $\mathcal{C}$ has prompt $p$ and server $\mathcal{S}$ has weights $W_{\mathsf{pr}}$. Both parties have $W_{\mathsf{pb}}$. Let $n$ be the number of tokens to be generated.

1. Both parties initialize an instance of $\mathcal{F}_{M_{\mathsf{pr}}}$ and the server $\mathcal{S}$ sends $W_{\mathsf{pr}}$ to $\mathcal{F}_{M_{\mathsf{pr}}}$.

2. The client locally evaluates the public part of the model on its prompt to get the hidden state for the prompt $h \leftarrow M_{\mathsf{pb}}.\mathsf{evaluate}(W_{\mathsf{pb}}, p)$, and sends $h$ to $\mathcal{F}_{M_{\mathsf{pr}}}$. Note that this is the input that $M_{\mathsf{pr}}$ expects to perform pre-filling on the prompt.

3. $\mathcal{C}$ receives $y_1$ from $\mathcal{F}_{M_{\mathsf{pr}}}$.

4. For $i = 2, \ldots, n$:
   (a) $\mathcal{C}$ locally evaluates the public part of the model on its prompt to get $h \leftarrow M_{\mathsf{pb}}.\mathsf{evaluate}(W_{\mathsf{pb}}, y_{i-1})$, and sends $h$ to $\mathcal{F}_{M_{\mathsf{pr}}}$. Note that this is the input $M_{\mathsf{pr}}$ expects to update its context state with $y_{i-1}$.
   (b) $\mathcal{C}$ receives $y_i$ from $\mathcal{F}_{M_{\mathsf{pr}}}$.

5. $\mathcal{C}$ outputs $(y_1, \ldots, y_n)$.

---

Figure 7: Our secure inference protocol.

secure inference protocol, which makes calls to $\mathcal{F}_{M_{\mathsf{pr}}}$ and does the rest of the inference computation outside MPC, securely realizes $\mathcal{F}_M$. That is, its security is equivalent to performing the entire inference computation within MPC. Note how this security argument is only concerned with the API provided by $\mathcal{F}$ and does not need to get into the proof specifics of each inference stage. Our proof strategy follows the standard simulation-based proof paradigm in the hybrid model (Canetti, 2000; Goldreich et al., 1987; Lindell, 2017) where a protocol $\Pi$ securely realizes an ideal functionality $\mathcal{F}$ if whatever an adversary $\mathcal{A}$ can learn about the private inputs of honest parties from $\Pi$ can also be learned by interacting with $\mathcal{F}$ which is secure by definition. This is proved by constructing a simulator Sim that can simulate the adversary's view in $\Pi$ by only interacting with the adversary $\mathcal{A}$ and $\mathcal{F}$. In Fig. 8, we describe a simulator for MARILL 's secure inference protocol (Fig. 7) which rigorously proves the following theorem. The proof follows trivially given the simulator.

**Theorem 1.** *In the presence of a semi-honest adversary, the protocol in Fig. 7 securely realizes $\mathcal{F}_M$ in the $\mathcal{F}_{M_{\mathsf{pr}}}$-hybrid model where $M$ and $M_{\mathsf{pr}}$ are defined above.*

---

**Simulator for MARILL's Secure Inference Protocol**

The simulator Sim internally runs the adversary $\mathcal{A}$, has access to its input prompt $p$ (since $\mathcal{A}$ is semi-honest), interacts with ideal functionality $\mathcal{F}_M$ on behalf of the party controlled by the adversary, and simulates $\mathcal{F}_{M_{\mathsf{pr}}}$ in the ideal-world.

If client $\mathcal{C}$ is corrupted:

1. Sim sends prompt $p$ to $\mathcal{F}_M$ and receives $y_1$ from it.

2. As $\mathcal{F}_{M_{\mathsf{pr}}}$, Sim receives $h$ from $\mathcal{A}$, ignores it, and sends $y_1$ to $\mathcal{A}$ as the output.

3. For $i = 2, \ldots, n$:
   (a) Sim sends $y_{i-1}$ to $\mathcal{F}_M$ and receives $y_i$ from it.
   (b) As $\mathcal{F}_{M_{\mathsf{pr}}}$, Sim receives $h$ from $\mathcal{A}$, ignores it, and sends $y_i$ to $\mathcal{A}$ as the output.

If server $\mathcal{S}$ is corrupted:

1. Receive model weights $W_{\mathsf{pr}}$ from $\mathcal{A}$, append it to the public weights $W_{\mathsf{pb}}$ to get $W = W_{\mathsf{pb}}\|W_{\mathsf{pr}}$ and forward $W$ to $\mathcal{F}_M$. There is nothing else to simulate since the server does not receive any messages in our protocol in the $\mathcal{F}$-hybrid model.

---

Figure 8: Simulator for MARILL's secure inference protocol.

# F MALICIOUS SECURITY

Our work is not limited to a semi-honest adversary and can also support a malicious adversary that deviates from the protocol arbitrarily. Given a maliciously-secure protocol, our work inherits malicious security against the server directly as the server does not have any additional capabilities in our system. The simulator for a corrupted server also remains the same. Security against client needs careful assessment because the client in our system inputs a hidden state (output of a transformer layer), as opposed to a sequence of tokens in traditional secure LLM inference. This does not impact semi-honest security because the client will follow the protocol and input the right hidden state. However, a malicious client can input a state that doesn't correspond to any sequence of input tokens[4] to potentially learn the model weights, or input a different token from what was generated to deviate the generation process. This issue can be fixed by making the following changes to the protocol:

- In step 2, the client additionally provides a zero-knowledge proof-of-knowledge (ZKPoK) (Goldwasser et al., 1985) proving that the hidden state it is secret-sharing corresponds to an actual sequence of tokens of the appropriate length.

- The secure inference protocol will output the token as well as a hiding commitment and its randomness to the client. Now, when the client will secret-share the hidden state for the latest token $y_{i-1}$ in step 4a, it'll additionally provide a ZKPoK proving that this state is consistent with the commitment received during the previous token generation.

- If either proof fails, the protocol will be aborted.

To complete the argument for malicious security, the simulator will be updated as follows:

- Since the adversary is now malicious, the simulator does not have direct access to its input. Instead, the simulator will receive ZKPoK proofs in addition to the hidden states from the adversary $\mathcal{A}$. It will extract the adversary's input from these proofs. The rest of the simulation follows exactly the same way.

# G DISTILLATION

The modifications we make to the model for MPC-minimization change its learned knowledge during pre-training, and simply fine-tuning it leads to a large accuracy loss. To bridge this accuracy gap, we turn to knowledge distillation (KD) (Hinton et al., 2015) in this work.

Fig. 1 summarizes our distillation workflow. First, we take the pre-trained model and apply the transformations that lead to an MPC-minimized architecture; the model thus obtained is the *student*. Then, we take the pre-trained model and fully fine-tune it to get the *teacher model*, representing the performance baseline we want to match. Finally, we use KD to ease the fine-tuning of the student model by matching its intermediate states with the teacher model. The student model thus obtained can then be used for secure inference.

For layer-freezing and LoRA, we have a one-shot distillation procedure because they preserve the pre-trained knowledge. Head-merging, on the other hand, requires a two-stage distillation process, similar in spirit to the strategy from MPCFormer (Li et al., 2023b). Now, we describe the two stages of distillation. The configurations without head-merging only perform the second stage.

1. **Stage I - Attention and Hidden States KD**: to accommodate head-merging, we match the student and teacher outputs of MHA in each (trainable or private) transformer layer using the following loss function: $\mathcal{L}_{\text{attn}} = \sum_{i=fN}^{N} \mathsf{MSE}(\mathbf{a}_i^S, \mathbf{a}_i^T)$, where $\mathbf{a}_i^S$ and $\mathbf{a}_i^T$ are the MHA outputs in the $i$-th transformer layer of the student and teacher, respectively, $f$ is the fraction of layers fine-tuned during training, and $N$ is the number of transformer layers. Similarly, we compute the distillation loss over hidden states after every (private) transformer layer: $\mathcal{L}_{\text{hidden}} = \sum_{i=fN}^{N} \mathsf{MSE}(\mathbf{h}_i^S, \mathbf{h}_i^T)$, where $\mathbf{h}_i^S$ and $\mathbf{h}_i^T$ are the hidden layer outputs in the $i$-th transformer layer of the student and

---

[4]The possible input token combinations are exponentially larger than the possible hidden states, even concretely at sequence lengths as small as $b = 6$, but we do not know if transformer layers represent an onto function.

teacher, respectively. For all experiments, we adopt coefficients $\alpha_{\text{attn}}$ and $\alpha_{\text{hidden}}$ for these two losses, and set them to $\alpha_{\text{attn}} = 0.1, \alpha_{\text{hidden}} = 5.0$. We choose this value so that the two losses have similar magnitude, and we empirically observe that this brings the best accuracy. We skip this stage in experiments that do not use head-merging.

2. **Stage II - Logits KD**: following stage I distillation, we employ supervised KD (Hinton et al., 2015; Sanh et al., 2019) to match the student's token-level probability distribution (or logits) with that of the teacher. We use forward KL divergence (KLD) to measure the similarity of the distributions (Agarwal et al., 2024). In addition to the distillation loss, we also minimize the negative log-likelihood (NLL) of the student's output on labels from the fine-tuning dataset. Overall, we use the following loss function in this stage: $\mathcal{L}_{\text{logits}} = \alpha_{\text{KLD}} \cdot \text{KLD}(\mathbf{z}^S, \mathbf{z}^T) + \alpha_{\text{NLL}} \cdot \text{NLL}(\mathbf{z}^S, y)$, where $\mathbf{z}^S$ and $\mathbf{z}^T$ are the logits of the student and the teacher model, resp., $y$ is the label from the fine-tuning dataset, and $\alpha_{\text{KLD}}$ and $\alpha_{\text{NLL}}$ are scalar weights for the KLD and NLL terms, respectively. For all experiments, we set $\alpha_{\text{KLD}} = 0.5, \alpha_{\text{NLL}} = 0.5$.

**Combining head-merging with other techniques.** When using head-merging independently, we initialize the student weights with that of the teacher, perform a head similarity analysis on the teacher, and then perform the two-stages of distillation. When head-merging is combined with layer-freezing, we perform the same procedure, except we replace teacher weights with the weights of the layer-freezing fine-tuned student.

**Other experiments.** Head-pruning and MPC-friendly approximations follow the same recipe as head-merging and require two-stage distillation. When combining MPC-friendly approximations with head-merging, we introduce them at the same time before stage I distillation.

## H  MARILL CONFIGURATION PER SECURE INFERENCE PROTOCOL

MARILL's techniques target various potential bottlenecks that occur in secure inference protocols. In this section, we discuss which combination of techniques is the most suitable for a given secure inference protocol.

- If the protocol is bottlenecked on arithmetic operations, one should use LoRA because it provides an asymptotic reduction in these operations[5].

- If the protocol is bottlenecked by non-arithmetic operations, consider the sequence length of the inference task. If the sequence length is large, prefilling will dominate the overall cost and self-attention will be the bottleneck. Head-merging will reduce all the non-arithmetic operations in self-attention and provide significant runtime and communication improvements. If the sequence length is small, decoding is likely to dominate the cost, and LoRA will present better runtime improvements.

- If there is no specific bottleneck, use layer-freezing and it will reduce overheads irrespective of the cost profile of the underlying protocol. For half the layer frozen, layer-freezing alone offers $2\times$ improvements across all inference scenarios and protocols. Otherwise, first apply one of the other two techniques, and then add layer-freezing on top.

## I  DETAILED HYPERPARAMETERS FOR EXPERIMENTS

We performed a best-effort hyperparameter optimization under our compute budget by varying the number of training epochs and batch sizes while keeping the other hyperparamters the same across experiments for a given benchmark. Table 3 reports the best configuration we found for each experiment. We use the same configuration for the ablations, i.e., layer-pruning uses the same hyperparameters as layer-freezing, and head-pruning uses the same parameters as head-merging. Experiments combining 2ReLU with MARILL (Table 1a) use the same parameters as the corresponding MARILL experiments without 2ReLU.

---

[5]Integer additions are also arithmetic but they have relatively negligible cost and can thus be ignored, leaving integer multiplications as the only arithmetic operation.

Table 2: Secure inference performance vs fraction of layers fine-tuned $f$.

| Setting | $f = 26/26$ | $f = 13/26$ | $f = 9/26$ | $f = 6/26$ | $f = 5/26$ | $f = 4/26$ |
|---|---|---|---|---|---|---|
| Prefilling Time | | | | | | |
| 2PC | $1.0\times$ | $2.1\times$ | $2.9\times$ | $4.3\times$ | $5.1\times$ | $6.3\times$ |
| 3PC | $1.1\times$ | $2.1\times$ | $3.1\times$ | $4.6\times$ | $5.5\times$ | $6.9\times$ |
| 2PC-Dealer | $1.0\times$ | $2.0\times$ | $2.9\times$ | $4.3\times$ | $5.1\times$ | $6.4\times$ |
| Prefilling Comm | | | | | | |
| 2PC | $1.0\times$ | $2.0\times$ | $2.9\times$ | $4.3\times$ | $5.2\times$ | $6.4\times$ |
| 3PC | $1.0\times$ | $2.0\times$ | $2.9\times$ | $4.3\times$ | $5.2\times$ | $6.5\times$ |
| 2PC-Dealer | $1.0\times$ | $2.0\times$ | $2.9\times$ | $4.3\times$ | $5.2\times$ | $6.4\times$ |
| Decoding Time | | | | | | |
| 2PC | $1.0\times$ | $2.0\times$ | $2.8\times$ | $4.1\times$ | $4.9\times$ | $5.9\times$ |
| 3PC | $1.0\times$ | $2.0\times$ | $2.8\times$ | $4.0\times$ | $4.7\times$ | $5.7\times$ |
| 2PC-Dealer | $1.0\times$ | $2.0\times$ | $2.8\times$ | $4.0\times$ | $4.7\times$ | $5.7\times$ |
| Decoding Comm | | | | | | |
| 2PC | $1.0\times$ | $2.0\times$ | $2.8\times$ | $4.3\times$ | $5.1\times$ | $6.1\times$ |
| 3PC | $1.0\times$ | $2.0\times$ | $2.8\times$ | $4.3\times$ | $4.9\times$ | $6.4\times$ |
| 2PC-Dealer | $1.0\times$ | $2.0\times$ | $2.8\times$ | $4.1\times$ | $4.8\times$ | $5.8\times$ |

Table 3: Batch size and number of epochs for all experiments.

| | MTBench epochs | bsz | HumanEval epochs | bsz | WMT22 epochs | bsz |
|---|---|---|---|---|---|---|
| Fine-tuned | 3 | 128 | 2 | 128 | 1.5 | 128 |
| LF | 5 | 128 | 4 | 64 | 1.5 | 128 |
| LoRA/LF+LoRA | 5 | 8 | 4 | 8 | 1.5 | 128 |
| HM/LF+HM - Stage 1 | 3 | 8 | 2 | 64 | 1.5 | 128 |
| HM/LF+HM - Stage 2 | 5 | 128 | 2 | 64 | 1 | 128 |

## J MARILL SECURE INFERENCE PERFORMANCE OVER SIGMA

We evaluated SIGMA (Gupta et al., 2024) on three NVIDIA A100 GPUs linked by a LAN connection (see setup in § 6). SIGMA's implementation does not include decoding, so we only evaluated MARILL's improvements for the prefilling stage. For this evaluation, we considered the LF=0.5 and HM=4 configuration of MARILL as the non-arithmetic operations are the bottleneck in SIGMA, and found that MARILL improved SIGMA's runtime and communication by $3.2\times$ and $3.3\times$, respectively.

## K MARILL SECURE INFERENCE PERFORMANCE OVER WAN

We conduct the same experiment from Fig. 4 on a WAN connection with 400 Mbps bandwidth and 40 ms latency (emulated using Linux traffic control `tc`). Table 4 summarizes these results. Here P and D denote prefilling and decoding runtime, respectively; we do not report communication because it remains the same as in the LAN setting (Fig. 4). It is evident that layer-freezing and head-merging have similar improvements over the WAN and LAN settings. On the other hand, LoRA improvements are smaller because network costs dominate the WAN runtime, which LoRA does not improve. These results show that MARILL also improves the secure inference costs in network-constrained scenarios.

## L LAYER FREEZING PERFORMANCE ABLATION

We perform an ablation study of the layer freezing technique (Table 5) on the MTBench benchmark (Zheng et al., 2024). The results show that the MTBench score remains relatively consistent up to LF=0.5 (13 layers frozen out of 26). However, beyond this point, there is an almost linear decline in score as fewer layers are fine-tuned. Based on these observations and a strong performance

Table 4: MARILL's secure inference improvement over a WAN network. See Fig. 4 caption for details of the experiment. P and D denote the prefilling and decoding runtime respectively.

| Variant | P-2PC | P-3PC | P-2PC-Dealer | D-2PC | D-3PC | D-2PC-Dealer |
|---|---|---|---|---|---|---|
| LF=0.5 | 2× | 2.1× | 2× | 2× | 2× | 2× |
| HM=4 | 1.2× | 2.4× | 2.2× | 1.1× | 1.1× | 1× |
| LoRA=64 | 2.2× | 1× | 1× | 1.1× | 1× | 1.2× |
| LF+HM/LoRA | 4.5× | 5.1× | 4.3× | 2.2× | 2.2× | 2.4× |

Table 5: Layer freezing ablation on MTBench (Zheng et al., 2024).

| # layers fine-tuned | MTBench Score |
|---|---|
| LF=26/26 | 5.02 |
| LF=22/26 | 4.95 |
| LF=18/26 | 4.97 |
| LF=13/26 | 4.77 |
| LF=8/26 | 3.39 |
| LF=4/26 | 2.21 |

of this configuration on other benchmarks, we chose to freeze half the layers (LF=0.5) across our experiments to strike an effective balance between accuracy and secure inference cost.

## M  LIMITATIONS

**Availability of open-source pre-trained model.** In this work, we introduce a novel paradigm that shows how the publicly available weights of an open-source pre-trained model can be leveraged to accelerate secure inference. This makes sense in many settings because the provider doesn't have to go through a very expensive pre-training process, and the best open-source models are among the best models out there Chiang et al. (2024); Liu et al. (2024); Yan et al. (2024); Wang et al. (2023). However, there could be domains that require specialized knowledge which does not benefit from the pre-trained knowledge of the available open-source models. In such cases, the provider has to pre-train their own model, and layer-freezing and LoRA improvements will no longer apply. We note that if there is significant relevant public data available for that domain, the provider also has the option to open-source its own pre-trained model to leverage our techniques.

**Delegation setting.** In this work, we focus on the secure inference threat models considered by prior work. These works assume that client is one of the MPC participants, and thus, having it evaluate a part of the network locally with layer-freezing actually reduces its overhead. This is because the MPC overhead on each participant is orders higher than plaintext inference Gupta et al. (2024); Li et al. (2023b). However, one could also imagine a *weaker threat model* for all of these settings where the client does not participate in the MPC at all. Rather, an additional server is introduced to the MPC with the *additional trust assumption* that it will not collude with the other servers involved in the MPC. In this case, our layer freezing technique is indeed adding additional overhead on the client, which might not be acceptable in some cases.

## N  SOCIAL IMPACT

This paper presents work that enables privacy-preserving inference, where both the user's input as well as the service provider's model weights stay private. While user privacy is needed in many applications and desirable in general, there is a potential concern of model misuse through malicious user prompts. This is not a fundamental issue though, as the checks that the services perform today on user prompts can also be performed within MPC without revealing them to the service provider. Alternatively, at the cost of additional client overhead, the client could be asked to create a zero-knowledge proof (Goldwasser et al., 1985) proving that its input satisfies some criteria.

