# OpenReview forum: "MPC-Minimized Secure LLM Inference"
_ICLR.cc/2025/Conference — Submitted to ICLR 2025_

### Official Review · Reviewer_LtzT · 2024-10-21

**Soundness:** 3
**Presentation:** 3
**Contribution:** 3
**Rating:** 8
**Confidence:** 4

**Summary:**

Inference over LLMs  has risks to leak private information such as prompts while LLMs are prevalent in many downstream AI tasks nowaday. MPC-based approaches have been well studied in the literature to tackle the issue. However, due to expensive overheads incurred by MPC, this work focuses on architectural changes to reduce expensive MPC operations to make the inference of LLMs efficient while preserving privacy.

**Strengths:**

1. This work innovatively introduces layer freezing and LoRA techniques from plaintext inference to secure inference. Especially it reduces bottlenecked matmul dimensions in MPC-based inference.
2. This work well describes threat models in secure inference of LLMs and explains why each component can be optimized with considering potential attacks. Also, it explains why some parts cannot benefit from optimization by nature of MPC, such as mixture of weights.

**Weaknesses:**

1. In Section 5.1, authors do not explain the criteria or threshold of the frozen fraction f.
2. Since only 2PC-Dealer supports GPU acceleration, the comparison to other MPC approaches with CPU only looks unfair due to the hardware difference. If we want to do apple-to-apple comparison, it is better to make 2PC-Dealer run with CPU only, too.

**Questions:**

1. In the real-world enterprise-level LLMs, the system usually takes millions of requests at a short time. So, instead of 2-PC or 3-PC setting, have you considered the extension of this work to n-PC setting or distributed setting?
2. According to Weakness 1, if the faction f is too large or too small, what will it happen?
3. Can you provide more information about the difference between head-pruning and proposed head-merging.

---

> ### Author Response · Authors · 2024-11-16
>
> Thank you for your time and the encouraging feedback!
>
> > In Section 5.1, authors do not explain the criteria or threshold of the frozen fraction f.
> > According to Weakness 1, if the faction f is too large or too small, what will it happen?
>
> We selected the layer freezing configuration (LF=0.5) based on an ablation study we performed on the MTBench benchmark (see Appendix K and Table 5 in the paper). The results show that score remains relatively consistent up to LF=0.5 (13 layers frozen out of 26). However, beyond this point, there is an almost linear decline in score as fewer layers are fine-tuned.
> Based on these observations and a strong performance of this configuration on other benchmarks, we chose to freeze half the layers (LF=0.5) across our experiments to strike an effective balance between accuracy and secure inference cost. We’ll explicitly clarify this in the evaluation section.
>
> > Since only 2PC-Dealer supports GPU acceleration, the comparison to other MPC approaches with CPU only looks unfair due to the hardware difference. If we want to do apple-to-apple comparison, it is better to make 2PC-Dealer run with CPU only, too.
>
> We compared with the 2PC-Dealer framework with GPU acceleration because we wanted to demonstrate Marill’s improvements over the most efficient configuration of this framework. It is important to note that the communication improvements over GPU will be the same over CPU, and communication is typically the bottleneck for 2PC-Dealer secure inference.
> Additionally, the runtime improvements on 2PC-Dealer and 3PC framework are similar, even though the former is run on GPU and the latter on CPU, indicating that Marill’s performance improvements are consistent across hardware configurations.
>
> > In the real-world enterprise-level LLMs, the system usually takes millions of requests at a short time. So, instead of 2-PC or 3-PC setting, have you considered the extension of this work to n-PC setting or distributed setting?
>
> In MPC, adding more parties typically makes the computation slower, unless you introduce a trust assumption for the additional parties. This is why 2PC+Dealer and 3PC settings are more efficient than 2PC – they have a weaker security guarantee due to this trust assumption.
> To horizontally scale secure inference, in 2PC, if you have $n$ servers, you can run $n$ parallel queries by distributing them across those servers.
> 3PC and 2PC-Dealer require an additional helper party other than the client and the server, and if this helper party can horizontally scale the same way, then these settings can also scale similarly.
>
> > Can you provide more information about the difference between head-pruning and proposed head-merging.
>
> Both head-pruning and head-merging are similar in that they aim to reduce the number of heads, but the key difference is that head-merging achieves that while preserving all the parameters of the foundational model. In contrast, head-pruning simply removes certain heads from the model along with their parameters.
> This distinction allows head-merging to achieve better ML performance than head-pruning (see section 6.4 in the paper).
> Furthermore, while head-merging is not useful for plaintext inference because the FLOPs remain the same, we show in Table 1b that it offers similar secure inference performance as head pruning due to the unique performance characteristics of MPC.

---

> > ### Comment · Reviewer_LtzT · 2024-11-30
> >
> > Thanks for your explanation and it clears all my questions! Great work!

---

### Official Review · Reviewer_RXhi · 2024-11-02

**Soundness:** 2
**Presentation:** 2
**Contribution:** 2
**Rating:** 5
**Confidence:** 5

**Summary:**

The paper presents a secure inference framework that combines MPC (Multi-Party Computation) and LLM, aiming to address privacy issues in LLM-based inference services by minimizing the use of MPC during secure inference. The proposed method reduces the number of costly operations required during MPC by introducing high-level architectural changes during the fine-tuning process.

**Strengths:**

The paper describes three techniques to minimize the use of MPC: Layer Freezing, Low-Rank Adaptation (LoRA), and Head Merging. These techniques reduce the cost of MPC in secure inference.

**Weaknesses:**

1-The paper's innovation is limited. The proposed method does not include improvements to the MPC protocol itself but focuses on different partitioning and head merging methods in the fine-tuning network structure. Regarding the Layer Freezing method in the partitioning approach, is there any reference basis for selecting how many layers to freeze for different LLMs to achieve a balance between accuracy and timeliness?

2-Insufficient comparison of accuracy in experimental results. The paper does not compare the accuracy of other methods that combine MPC and LLM. Please supplement with more relevant experiments.

**Questions:**

Please see Weaknesses.

---

> ### Author Response · Authors · 2024-11-16
>
> > The paper's innovation is limited. The proposed method does not include improvements to the MPC protocol itself but focuses on different partitioning and head merging methods in the fine-tuning network structure.
>
> Thank you for the feedback! We agree that our work does not focus on modifying MPC backends, and we want to emphasize that this is intentional and central to our contribution. Our goal is to make changes to the machine learning (ML) pipeline that outputs the inference models so that it is significantly cheaper to perform secure inference on these models. A key strength of our approach is that by making MPC-tailored changes to the ML itself, we are able to offer secure inference improvements irrespective of the underlying MPC protocol.
>
> **Within the context of secure inference, all three of our techniques are novel**:
> * **Layer freezing and LoRA**: while these techniques were originally introduced to accelerate fine-tuning, our work is the first to leverage these techniques in the secure inference setting. We identify how to adapt these techniques to both preserve security and improve efficiency, and present a novel application of these ideas.
> * **Head merging**: this technique is entirely novel and uniquely suited to secure inference. It relies on the unique performance characteristics of MPC to improve secure inference cost while preserving FLOPs; note that the plaintext inference cost will stay the same.
>
> > Regarding the Layer Freezing method in the partitioning approach, is there any reference basis for selecting how many layers to freeze for different LLMs to achieve a balance between accuracy and timeliness?
>
> We selected the layer freezing configuration (LF=0.5) based on an ablation study we performed on the MTBench benchmark (see Appendix K and Table 5 in the paper). The results show that score remains relatively consistent up to LF=0.5 (13 layers frozen out of 26). However, beyond this point, there is an almost linear decline in score as fewer layers are fine-tuned.
> Based on these observations and a strong performance of this configuration on other benchmarks, we chose to freeze half the layers (LF=0.5) across our experiments to strike an effective balance between accuracy and secure inference cost. We’ll explicitly clarify this in the evaluation section.
>
> > 2-Insufficient comparison of accuracy in experimental results. The paper does not compare the accuracy of other methods that combine MPC and LLM. Please supplement with more relevant experiments.
>
> Thank you for raising this concern. To the best of our knowledge, the prior works that combine MPC and LLM primarily fall in the MPC-friendly approximations category discussed in the related work section of the paper.
> **Marill is complementary to these works** as it makes high-level changes to the architecture, as opposed to the underlying operations. We also show in section 6.3 that these approximations can be combined with Marill to yield further performance improvements.
> We did not include most prior works in our experiments because they rely on neural architecture search (NAS), where the model architecture is dependent on private training data. This dependency introduces additional information leakage beyond the standard secure inference guarantees, which conflicts with the strict security requirements we aim to preserve. In contrast, Marill’s architecture is statically configured, independent of the training data, and maintains the same security properties as standard secure inference. We’ll clarify this explicitly in the related work section.
>
> If there are specific works we may have overlooked that do not fall into the MPC-friendly approximations or NAS categories, we would be happy to review and consider them for inclusion.

---

### Official Review · Reviewer_P2FW · 2024-11-02

**Soundness:** 4
**Presentation:** 2
**Contribution:** 2
**Rating:** 6
**Confidence:** 4

**Summary:**

This paper considers a setting where a public open-source model is fine-tuned on private data so that the resulting (private) model can be used for private inference. They key contributions is the design of a series of optimizations that make it so that the "private part" of the fine-tuned model is comparatively small w.r.t. the full model, and furthermore is located only towards the final layers of the architecture. This allows the client with the private query to "preprocess" most of the model computation in the clear, given that the first layers are all public. Then, the MPC protocol would only need to execute the final private layers.

The optimizations include (1) freezing many of the initial layers prior to fine-tuning, (2) merging some of the attention heads to minimize the impact of their evaluation under MPC, and (3) using LoRA to reduced the dimension of the involved matrices in MPC. They show that (different combinations of) these can lead to reasonably small degradation loss wr.t. full fine-tuning while leading to substantial efficiency gains in the MPC context.

**Strengths:**

The ideas considered are sound, and the authors provide an implementation for reproducibility (which, disclaimer, I didn't verify myself). The idea of head-merging is, to the best of my knowledge, novel. It is relevant as it seems to work well in terms of accuracy, but it boosts the efficiency of MPC noticeably. The experiments are comprehensive and thorough. They are well described and enable good reproducibility.

Overall the paper is well written.

**Weaknesses:**

I am not sure the problem is very well motivated. This is not about running inference obliviously, which is very well motivated. This is is what most (if not all) of the cited prior works consider: a model owner wants to keep their model private and a client wants to keep the query hidden, while still learning the inference result. This works starts from the premise that the model that the client may want to query is actually a fine-tuned version of an open-source model, the the data for fine-tuning is private. I am not sure about in which category this falls and what types of use-cases it solves. I will elaborate on this in my questions below. However, in general, I don't think the setting is clear and I am unconvinced whether it is really that relevant (in particular, since the paper cites as baselines works that study a fundamentally different problem).

The claimed proof in the appendix is very high level and feels more like a "checkbox" rather that anything that is supposed to shed any light. In particular, there are several details unspecified which an actual proof would need to deal with. For instance, the functionality is inherently defined over real values, but all of the protocols cited in the paper operate over fixed point values. There's no way then to instantiate the functionality as written. Furthermore, most of the cited protocols use probabilistic truncation for fixed-point arithmetic, which is not provably secure (https://www.usenix.org/conference/usenixsecurity23/presentation/li-yun).

**Questions:**

The use-case considered here somewhat differs with what has appeared in prior works. Considering public models first posses the question: what does it mean to "fine-tune on private data"? who owns this data? this makes the most sense only if the data is held by a collection of parties, since if it's held by only one party, what is preventing this party from performing the fine-tuning themselves? However, in this case, the fine-tuning would this an MPC protocol among who? Who runs the secure computation? I think these details are unclear and are important to understand the context, and the associated inefficiencies. In particular, at the end of the fine-tuning the model is "private" (well, part of it), so this probably means shared among at least two parties... who is the client then who performs the query at inference time? If it's an additional party than this has become a 3-party protocol. These details matter and they are left unspecified.

Why do you cite [73] as the SOTA for 2PC+Dealer? Earlier in the text you refer to Sigma as such SOTA (which makes sense).

NOTE: SIGMA appeared at PoPETs 2024. Check this and other preprint references.

---

> ### Author Response · Authors · 2024-11-16
>
> > Considering public models first posses the question: what does it mean to "fine-tune on private data"? who owns this data? this makes the most sense only if the data is held by a collection of parties, since if it's held by only one party, what is preventing this party from performing the fine-tuning themselves? However, in this case, the fine-tuning would this an MPC protocol among who? Who runs the secure computation? In particular, at the end of the fine-tuning the model is "private" (well, part of it), so this probably means shared among at least two parties... who is the client then who performs the query at inference time?
>
> Thank you for your feedback! It seems that these questions arise from a misunderstanding about our work’s objective. We will clarify the objective more explicitly in the introduction, and hope that the following clarifications will address your concerns.
>
> Marill operates in a setting where the foundational model is open-source and the fine-tuning data (which is sensitive or proprietary) is known entirely by a single service provider.
>
> In this setting:
> - **Fine-tuning is performed locally on the service provider, with no involvement of MPC** (see Fig. 1 caption).
> - The goal is to leverage the open-source model to achieve better ML performance than a model that is trained on the private data alone. This aligns with the standard workflows of leveraging advanced foundational models.
>
> **The resulting model can then be served to any client through a secure inference API**.
> Using traditional secure inference, the service provider will pay the secure inference cost of the entire foundational model.
> With Marill, only a part of the foundational model needs to evaluated within MPC during secure inference without compromising security, and this results in the secure inference performance improvements we report.
>
> > The claimed proof in the appendix is very high level and feels more like a "checkbox" rather that anything that is supposed to shed any light. In particular, there are several details unspecified which an actual proof would need to deal with. For instance, the functionality is inherently defined over real values, but all of the protocols cited in the paper operate over fixed point values. There's no way then to instantiate the functionality as written. Furthermore, most of the cited protocols use probabilistic truncation for fixed-point arithmetic, which is not provably secure.
>
> We hope that the above clarification on Marill’s setting helps address some of your concerns about the rigor of the proof.
> Besides the issues you’ve highlighted (addressed below), we are happy to hear any additional concerns you may have.
>
> We agree that we should clarify that the ideal functionality $F_A$ runs the fixed-point converted model because that is what all prior works support. Each work also has a slightly different way to perform fixed-point approximations of real values and we can also make that clear. So we will basically prove the following statement:
>
> “Given a secure inference protocol $P$ that securely realizes an ideal functionality $F_A$, where $F$ applies $P$’s fixed-point approximations to the model architecture $A$, Marill, by making black-box calls to $P$ also securely realizes $F_A$”
>
> Given this change, there are secure inference protocols that securely realize some $F_A$ and we can clarify that this proof only applies to those that do.
>
> While it is true that most of the cited protocols use probabilistic truncation for fixed-point arithmetic, we note that Marill does not introduce any changes to the underlying arithmetic operations as it only makes black-box calls to a secure inference protocol. Thus, the security guarantees remain consistent with those established in prior works. We will explicitly state this limitation and reference the potential vulnerability [Li et al.].
>
> > Why do you cite [73] as the SOTA for 2PC+Dealer? Earlier in the text you refer to Sigma as such SOTA (which makes sense).
>
> Thank you for your question! We cite CrypTen [73] as a SOTA secure inference framework for 2PC-Dealer because it remains widely adopted in both academia ([1,2,3,4]) and industry. This is largely due to its user-friendly frontend, which seamlessly integrates with PyTorch, enabling secure inference directly on PyTorch code.
> At the same time, we also refer to SIGMA as a SOTA secure inference protocol for 2PC-Dealer because it offers the most efficient backend for secure inference.
>
> Marill is complementary to both CrypTen and SIGMA and improves their performance. We report improvements on CrypTen in section 6.1, and Marill’s LF=0.5 + HM=4 configuration improves SIGMA’s pre-filling runtime and communication efficiency by $3.2\times$ and $3.3\times$, respectively. We will include these results in the paper for additional clarity and evidence of Marill’s improvement.
>
> [1] SecFormer. ACL 2024.
> [2] SAL-ViT. ICCV’23.
> [3] RNA-ViT. ICCAD’23.
> [4] MPCFORMER. ICLR 2023.

---

> > ### Comment · Reviewer_P2FW · 2024-11-18
> >
> > Thanks for your response. Indeed, I didn't understand the setting well, and your explanation helps. The setting is a classical 2-party inference setting and your techniques show how the model owner can train (fine-tune) a "better" model that incurs less overhead in MPC.
> >
> > My comment on CrypTen was for the sake of future readers. To the best of my understanding CrypTen is not a protocol but an implementation/framework: Unlike Sigma, it doesn't contain any novel cryptographic ideas, so the notion of "SOTA" seems wrong here. In my opinion it creates the idea that the protocol itself is SOTA, compared to the points you said (implementation being easy to use, other works using it for experiments, etc.) In any case this is just writing.
> >
> > Regarding the rigor of the proof, I understand what you're trying to emphasize: "Marill does not introduce any changes to the underlying arithmetic operations as it only makes black-box calls to a secure inference protocol". I agree with this. However, what triggered me precisely is that you aimed at including a "proof" where this was clearly not within the scope of your work. It feels very clearly as a "checkmark" and does not shed any light on any virtue or value of your work.
> >
> > Even with the changes you suggest, I still don't see the point of the proof or what you can prove in a first place. Furthermore, this is orthogonal to the fixed-point issue, which I brought up mostly to highlight that the proof is not even "complete" in a first place. The functionality is typically "Alice has a model, Bob has a query, Bob gets result and Alice learns nothing", and the papers you cite instantiate this. Now, however, you're changing the functionality: Alice has a model with some part that is public and some part that is private, and you're using a protocol to evaluate the private part. What is there to "prove"? My suggestion is to remove in its entirety any mention to proofs of cryptographic formality, and focus on the qualitative argument that you focus on the ML part.

---

> > > ### Author Response · Authors · 2024-11-19
> > >
> > > > However, what triggered me precisely is that you aimed at including a "proof" where this was clearly not within the scope of your work. It feels very clearly as a "checkmark" and does not shed any light on any virtue or value of your work.
> > > > Even with the changes you suggest, I still don't see the point of the proof or what you can prove in a first place. Furthermore, this is orthogonal to the fixed-point issue, which I brought up mostly to highlight that the proof is not even "complete" in a first place. The functionality is typically "Alice has a model, Bob has a query, Bob gets result and Alice learns nothing", and the papers you cite instantiate this. Now, however, you're changing the functionality: Alice has a model with some part that is public and some part that is private, and you're using a protocol to evaluate the private part. What is there to "prove"? My suggestion is to remove in its entirety any mention to proofs of cryptographic formality, and focus on the qualitative argument that you focus on the ML part.
> > >
> > > Thank you for raising the score and elaborating on your concerns regarding the proof.
> > > In this response, we will clarify why we included the proof and hopefully that will help justify why the proof was not meant as a checkmark.
> > >
> > > We agree that our proof is not providing any insights about the underlying secure inference protocol itself and its proof is beyond the scope of our work. However, we do introduce a public-private architecture, where a part of the model is public and a part is private, and with the proof, we intend to show that this does not introduce any security issues.
> > > The way we prove this is by showing that “running just the private part of the model within MPC is equivalent to running the whole model within MPC”.
> > > Note that this can not be proved for an alternate public-private architecture that introduces some leakage. For instance, an architecture where every other layer is private and you evaluate just the private part within MPC would have additional leakage from the intermediate layers.
> > >
> > > That said, we acknowledge that the proof might seem straightforward and unnecessary for readers familiar with these concepts. If you feel that it does not add significant value, we are open to replacing it with a discussion arguing the above points.

---

> > > > ### Author Response · Authors · 2024-11-26
> > > >
> > > > Dear Reviewer P2FW,
> > > >
> > > > We wanted to follow up on our previous comment regarding the proof in our draft. Do you think it adds sufficient value given our arguments, or should we replace it with a qualitative discussion instead? We’re finalizing the revision and plan to upload the updated draft by Nov 27, so your input would be greatly appreciated.
> > > >
> > > > Best regards, The Authors

---

> > > > > ### Comment · Reviewer_P2FW · 2024-11-26
> > > > >
> > > > > Thanks for following up.
> > > > >
> > > > > I think having a corrected proof in the Appendix with an annotation that addresses "nuanced questions" such as mine on precision, probabilistic truncation and whatnot is useful. I just would avoid using the proof as any sort of leverage.

---

### Author Response · Authors · 2024-11-28

Dear Reviewers,

We have uploaded a revised PDF of the paper addressing your feedback. Below is a summary of changes made (marked in blue in the PDF):

### Reviewer P2FW
---
- Clarified the subtleties of the proof, emphasized its straightforward nature, and outlined the scope of what is being proven.
- Clarified that Crypten is a popular 2PC-Dealer framework and SIGMA is the SOTA 2PC-Dealer protocol.
- Added SIGMA evaluation results.
- Clarified early on in the introduction that fine-tuning is performed locally at the service provider.
### Reviewer RXhi
---
- Qualified our claims in the introduction and added a discussion on the use of layer freezing and LoRA in prior TEE-based secure machine learning, and how it differs from our setting.
- Added more explicit references to layer freezing ablation and improved accuracy from head merging (HM=2) in section 6.2 on ML accuracy.
- Clarified why we don’t include NAS-based prior works in the experiments.
- Updated the figure caption to clarify that the client also knows the model architecture for the private model component input to MPC.
### Reviewer LtzT
---
- Added explicit reference to layer freezing ablation which explains the criteria for choosing LF=0.5.

We also made edits throughout the document to accommodate new content and enhance readability. We appreciate your valuable feedback and look forward to hearing your thoughts.

Best regards, The Authors

---

### Meta-Review · Area_Chair_7KEB · 2024-12-21

**Metareview:**

The submission proposes a framework that adapts LLM fine-tuning to improve secure multiparty computation (MPC) efficiency during inference. The paper shows substantial improvements in efficiency compared to full fine-tuning by using a combination of 3 methods: (i) layer freezing, (ii) Low-Rank Adaptation (LoRA), and (iii) head-merging (whereby multiple attention heads are merged into one).

The main downside of this submission is that its novelty is limited. In fact, layer freezing and LoRA are so widely used that they could be considered as natural baselines in this study as opposed to novel algorithmic contributions. On the other hand, while head-merging is a new contribution of this work, the paper would benefit from a more clear discussion (and ablations) of the benefits of head-merging compared to that of applying a combination of layer freezing and/or LoRA. In its present form, the paper does not meet the novelty bar for publication at ICLR.

**Additional Comments On Reviewer Discussion:**

The reviewers raised different concerns regarding this submission including:

1) The problem motivation
2) The proof in the appendix
3) The paper’s contribution
4) The experimental evaluation
5) The comparison to other MPC approaches

While the authors satisfactorily addressed concerns 1), 2), 4) and 5) in the discussion, the concern around 3) remains.

---

### Decision · Program_Chairs · 2025-01-22

Reject